# Usability Evaluation of the SmartWheeler through Qualitative and Quantitative Studies

**DOI:** 10.3390/s22155627

**Published:** 2022-07-27

**Authors:** Adina M. Panchea, Nathalie Todam Nguepnang, Dahlia Kairy, François Ferland

**Affiliations:** 1Laboratoire de Robotique Intelligente/Interactive/Intégrée/Interdisciplinaire, Institut Interdisciplinaire d’Innovation Technologique (3IT), Université de Sherbrooke, Sherbrooke, QC J1K 0A5, Canada; francois.ferland@usherbrooke.ca; 2Programme de Physiothérapie, École de Réadaptation Faculté de Médecine, Université de Montréal, Montreal, QC H3T 1J4, Canada; nathalie.todam.nguepnang@umontreal.ca (N.T.N.); dahlia.kairy@umontreal.ca (D.K.)

**Keywords:** assistive technology, powered wheelchair, mobility aids, intelligent wheelchairs

## Abstract

Background: Intelligent powered wheelchairs remain a popular research topic that can improve users’ quality of life. Although our multidisciplinary research team has put a lot of effort into adding features based on end-users needs and impairments since 2006, there are still open issues regarding the usability and functionalities of an intelligent powered wheelchair (IPW). Methods: For this reason, this research presents an experience with our IPW followed by a study in two parts: a quantitative one based on the System Usability Scale (SUS) questionnaire and a qualitative one through open questions regarding IPW functionalities with novice users, e.g., IPW non-users. These users never used an IPW before, but are users and aware of the impacts of the technology used in our IPW, being undergraduate to postdoctoral students and staff (faculty, lecturers, research engineers) at the Faculty of Engineering of Université de Sherbrooke. Results: The qualitative analyses identified different behaviours among the novice users. The quantitative analysis via SUS questionnaire done with novice users reports an “okay” rating (equivalent with a C grade or 68 SUS Score) for our IPW’s usability. Moreover, advantages and disadvantages opinions were gathered on the IPW as well as comments which can be used to improve the system. Conclusions: The results reported in these studies show that the system, e.g., IPW, was judged to be sufficiently usable and robust by novice users, with and without experience with the software used in developing the IPW.

## 1. Introduction

There are currently more than 1 billion disabled people in the world [1]. According to the *World Health Organization* (WHO), a disabled person is anyone who has “a problem in body function or structure, an activity limitation, has a difficulty in executing a task or action; with a participation restriction”. Moreover, 75 million people need a wheelchair on a daily basis. This represents 1% of the world’s population, which is twice Canada’s population. In 2012, Statistics Canada [2] reported that among the estimated 3,775,920 individuals with a disability, 8% used a wheeled mobility device, 10% reported an unmet need for an additional device: a power wheelchair (5%), a mobility scooter (4%) or a manual wheelchair (1%). While for non-users who needed a wheeled mobility device, the most common was a mobility scooter (60%), followed by a power wheelchair (21%) and a manual wheelchair (19%).

Powered wheelchairs (PW) were designed to improve the quality of life of people for which manual wheelchairs became difficult to use [3]. For similar reasons, intelligent power wheelchairs (IPWs) were developed to help the needs [4,5] of people with cognitive/motor/sensory impairment and their caregivers [6], whether it is due to disability or disease, by providing collision avoidance, navigation support and adapted control modes. IPWs are PWs in which computers and a collection of sensors are integrated to enable multimodal control systems and safe autonomous navigation, but also flexibility.

Since 2006 when work started on our IPW through the SmartWheeler project [7], many technological advancements were reported in the literature for IPWs, in which researchers propose hand gesture recognition [8] to control IPW movements, e.g., forwards/draw back/turn left/turn right/stop, or posture recognition [9] which can provide information on users behaviours and maybe unlock fatigue estimation and activity level assessment. There are also IPWs with tactile screen interfaces, and autonomous navigation while avoiding obstacles even in an unknown and dynamic environment [10,11] which can provide a certain level of autonomy to IPW users with good tactile sensing. Moreover, Ref. [12] developed and designed a LIDAR scanner for wheelchair applications as commercial ones seem to not satisfy requirements for IPW applications. Although the literature reports innovative and ambitious wheelchair technology, there is a need to evaluate their performance quality, safety and usability. For example, Ref. [11] argues that it is important to have a standardised test to evaluate autonomous wheelchairs, while [4,5] underline the importance of involving users in the ongoing development of IPWs after conducting a study involving interviews with users and caregivers, or with users in long term care [6].

In this study, the IPW utilised is the one reported in the SmartWheeler project [13] and consists of an autonomous wheelchair with collision avoidance capabilities and a tactile screen which we improved by adding a user-friendly interface to give the users means to communicate with it and better mechanical parts. The development of such touch screen interface was the result of feedback gathered from a previous study with users in a mall center [4] which indicated that the manual joystick is hard to physically be maintained for long periods of time. We report a quantitative and qualitative study in which the IPW is tested by non-end-users (called *novice* users in the following) in order to rapidly examine its usability and functionalities before involving end-users in real environments scenarios.

Testing an IPW’s usability or capturing people’s perceptions is not new: Viswanathan et al. [14] captured the user attitudes, needs and preferences towards IPWs using the Wizard-of-Oz method [15] of rapid prototyping without developing a fully functional prototype while collecting self-report ratings of satisfaction through the QUEST 2.0 survey [16], and of task load index through the NASA-TLX survey [17]. Ref. [14] results imply that familiarity and training with the system might impact user preference and usability. Moreover, a preliminary study [18] utilised both manual and powered wheelchair users to examine the usability of express buses to identify the effectiveness, efficiency and satisfaction of wheelchair boarding systems, reporting acceptable boundary while using the System Usability Scale (SUS) questionnaire. The SUS questionnaire was used within the IntellWheels project [19] with non-users to test the multimodal interface developed on an IW in order to prepare its complete test with users. The aim of this study’s strategy is to test the developments made by our team on the IPW based on the feedback gathered while using it with end-users collected in a previous experiment (e.g. the added tactile screen interface) and to rectify or/and adjust them, if necessary, before carrying out tests with the end-users. We believe that after years of development from a technological perspective, involving novice users with access to technology to test the IPW capabilities, e.g., manual/semi-autonomous/autonomous modes of control, can indicate the level of IPW’s usability and what should be improved before moving further and involve end-users. Moreover, the choice to conduct such a study with *novice* users was also due to the pandemic restrictions imposed in January 2021 when the experiments were conducted, which were not in favour of approaching IPW users for experiments.

Finally, the study aims at, first, qualifying: (a) the familiarisation with the IPW by novice users before executing a predefined course (following the tracks on the floor of one of our experimental spaces at Université de Sherbrooke) and (b) a sort of participant observation (https://research.utoronto.ca/participant-observation, accessed on 1 January 2021) in which the questions and discussions between the researcher which conducted the experiment and each participant emerge during the involvement of the participant. Secondly, (i) the IPW usability through an online questionnaire on Lime Survey (https://sondage.crir.ca/index.php?r=survey/index&sid=488847, accessed on 1 January 2021) consisting of the System Usability Scale (SUS) questionnaire [20] and (ii) the IPW’s functionality through four open-ended questions were quantified.

Briefly, our results show that by involving novice users, new ways of improvements in terms of functionalities have been reported. Moreover, in terms of usability the IPW is reported to fit the requirements for which it has been designed.

## 2. Materials and Methods

In this study, we used an evolution of the IPW that was used in the original SmartWheeler project [13]. The original goal of the SmartWheeler project was the minimisation of the physical and cognitive load required in controlling it.

The IPW is composed of a commercially available Sunrise Quickie Freestyle, to which we have added front right, front-left and back light detection and ranging (LIDARs), wheel odometers, an emergency button, a touch-sensitive graphical display and an onboard computer. The LIDARs and odometers are used for navigation and obstacle avoidance. The onboard computer interfaces with the wheelchair’s motor control board to provide autonomous navigational commands. All original hardware and electronic design were performed in-house by staff members and students at McGill’s Center for Intelligent Machines (CIM) and École Polytechnique de Montréal. Further hardware maintenance and software development were done at Université de Sherbrooke’s IntRoLab, notably on the interaction interface. The SmartWheeler used in this study uses the tactile/visual interface system as a communication way between the IPW and the user, and not only to provide visual feedback to the user regarding the state of the dialogue system as done previously [11,13]. Thereby, the touchscreen display and manual joystick (which comes with the commercial platform) are the main modes of communication with the user.

The touchscreen display is the new feature added as the result of the feedback provided by some users who found it difficult to hold the manual joystick in a stable position for a long time. The graphical interface proposed on the touchscreen display and reported on Figure 1 is designed using the Rviz (http://wiki.ros.org/rviz, accessed on 1 January 2021) graphical interface proposed by Robot Operation System (ROS) (https://ros.org, accessed on 1 January 2021), which is an open source set of libraries popular in robotics. The screen displays:On the right side, a virtual IPW in its environment (described by a map) along with the obstacles (reported in blue color in Figure 2) as detected by the LIDAR sensors. The environment is constructed and updated using simultaneous localization and mapping (SLAM).On the left side, the four interaction modes as described in the following.

The IPW can be controlled by the user with four interaction modes:Manual mode, using the joystick which comes along with the commercially available wheelchair platform and three other modes which can be activated via the touch screen as reported on Figure 1.Semi-autonomous mode, designed by our teams as a new feature added to the SmartWheeler, composed of:-A virtual joystick, as represented in Figure 2 upper left side, which allows the user to move the IPW in the desired direction by moving the black button in the circle and-An IPW-shaped-joystick, side, which is represented as an IPW shape shadow on the map, as represented on Figure 2 upper right, and allows the user to move the IPW in the desired direction by dragging the IPW’s shadow on the map;The difference between the two semi-autonomous modes consists in their design.Autonomous mode (represented on Figure 2) lower left and right, which consists, first, of indicating a desired destination on the map using a virtual representation of the IPW, and second, the wheelchair will autonomously generate the path to reach the desired destination.

**Figure 1 sensors-22-05627-f001:**
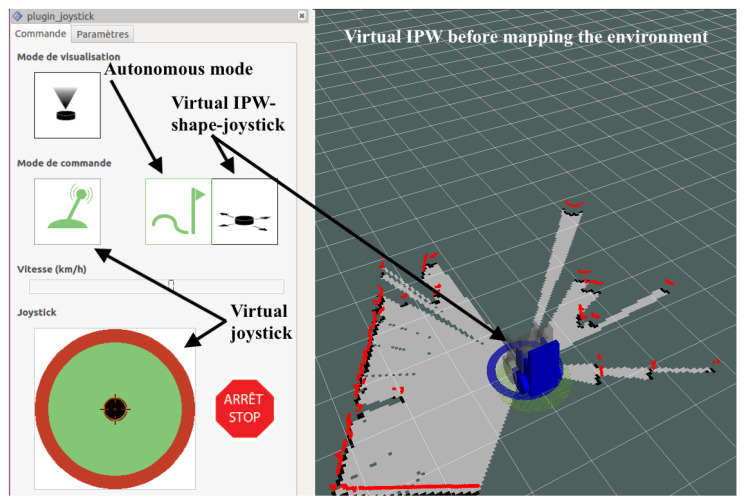
Graphical interface present on the IPW’s touch screen tablet when first starting the IPW.

**Figure 2 sensors-22-05627-f002:**
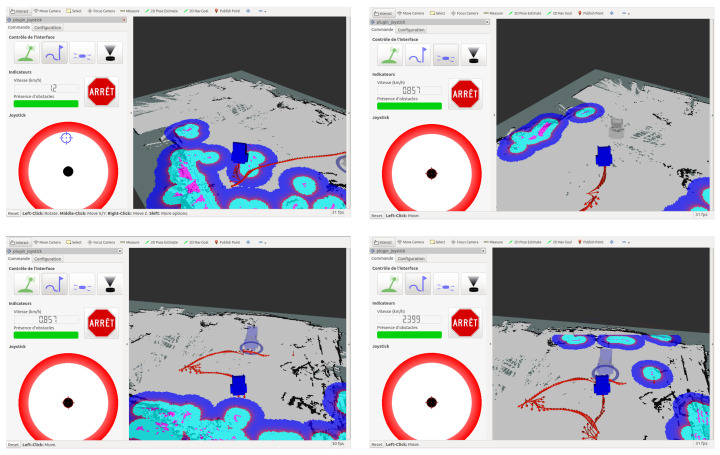
Graphical interface representing (**upper left**) the virtual joystick, (**upper right**) the IPW-shape-joystick and (**lower right** and **lower left**) the autonomous mode.

In order to validate the development carried out on the IPW and to correct problems that could appear due to the placement of the LiDAR sensors (for the detection of obstacles), the sensitivity of the touchscreen or the development of the semi-autonomous and autonomous operating modes, this study carries out an experiment with novice users on our IPW, as reported in the following. The study was approved by the Centre for Interdisciplinary Research in Rehabilitation of Greater Montréal (CRIR) Research Ethics Committee.

### 2.1. Participants and Recruitment Procedure

Using a convenience sample, eleven IPW novice participants were recruited for this study, as following: an email invitation to participate was sent to the mailing list of students, enrolled in all cycles of study from undergraduate to postdoctoral, and staff (faculty, lecturers, and research engineers) at Université de Sherbrooke. The email summarised our study. Interested persons contacted our research team by email, and a meeting was scheduled based on each participant’s availability. Moreover, the inclusion criteria were:(a)being a minimum of 18 years of age,(b)being a student or staff member of the Université de Sherbrooke (as access to the institute restrictions was restricted to faculty members during the COVID-19 lockdown) and(c)fluent in English or French, while the exclusion one was (a) to have a cognitive, visual or auditory deficit.

Each participant received, after the experiment, a compensatory allowance of CAD 20 for their participation.

### 2.2. Experimental Setup and Protocol

The purposes of using novice users in the proposed experimental are to:Analyze, identify and/or evaluate the control graphical interface, reported on the touch screen, as represented on Figure 1, for improvements and complementary and/or alternative interaction modalities;Identify the specifications retained in connection with the comfort improvement and preferably in the planning of the IPW driving.

To do so, participation in the experiment was meant to last approximately one and a half hours and could be held in one session or more depending on the availability of the novice participants.

The experimental room at the Interdisciplinary Institute for Technological Innovation (3IT) at Université de Sherbrooke used for the experiments in this study was mapped prior to welcoming the participants by a member of our team.

First, the researcher went through the consent form and if the participant agreed to sign it then the researcher presented the physical IPW and verbally described all its functionalities and operation modes. Secondly, the novice users were asked to complete the following tasks in the following order:**(1).** **Familiarise with the IPW** The objective was to become familiar with the IPW, meaning each participant manipulated all the options of the IPW while the team member was at their side to answer all their questions. For safety reasons, an emergency stop for the IPW was available on the IPW and a second one was available to the research member.**(2).** **Run a predefined path** Participants were asked to use the IPW to run a predefined path which consisted in following the green lines attached to the floor of the experimental room used for the experiments. The only indication given by the researcher was to follow the path as many times as they want but the only constraint was to use, at their convenience, one after the other on the same round or on separate rounds, the four interaction modes present on the IPW:manual mode via a physical joystick,semi-autonomous mode represented by (a) a virtual joystick present on the graphical interface and (b) a virtual IPW-shape joystick,and an autonomous mode, by indicating (dragging the virtual IPW present on the touch screen on the desired location).**(3).** **Filling out a questionnaire** At the end of the experiment, the participants were asked to complete an online questionnaire via Lime Survey to assess their ability to use the IPW and comments regarding its functionalities during the experiment.

### 2.3. Data Collection and Analysis

In order to capture data from the involvement of novice participants, the following methods were used:Qualitative method each participant was observed using the participant observation technique (https://research.utoronto.ca/participant-observation, accessed on 1 January 2021) by the researcher who conducted the experiment, in which the questions and discussions emerge during the involvement of the participant, during the familiarisation with the IPW by novice users before executing the predefined path (which consisted of following the green tracks on the floor of one of our experimental room at Université de Sherbrooke as represented on Figure 3).Quantitative method after performing the experiment the participants filled (i) a Lime Survey (https://sondage.crir.ca/index.php?r=survey/index&sid=488847, accessed on 1 January 2021) online questionnaire on the IPW usability, consisting of the System Usability Scale (SUS) questionnaire and answered to (ii) four open-ended questions related to IPW functionality developed by our research team.

The questionnaire used for the quantitative method was available on a desktop PC installed in the same room where the tests were performed.

The collected data were analysed as follows:Qualitative analyses The qualitative analyses, which in our study was of the participant observation type, involved the observer asking questions and discussing with the participant the function of the participant’s involvement. Moreover, this technique involves observing the participant’s behaviour in the experimental environment. Thereby, the data collected by the observer include an unstructured interview with the participants, notes based on their observations and interactions on how the participants performed the (1) Familiarise with the IPW step: in what order they tested the modes and the IPW, what and how many questions they had during this step, how much time they needed to complete the three tasks and the ease or difficulty they had when familiarizing with the device. This information was written by the observer, reported to the other authors and analysed after a group decision.Quantitative analyses The quantitative analyses is performed:-on the SUS questionnaire, as described in the following,-on the answers provided by the participants on the four open-ended questions related to IPW functionality.When the SUS is used, participants are asked to score the 10 items reported in Section A.1 with one of five responses that range from strongly agree to strongly disagree (Likert scale). It is a mixed-tone questionnaire in which the odd-numbered items have a positive tone and the even-numbered items have a negative tone.The first step in scoring on the SUS is to determine each item’s score contribution, which will range from 0 to 5. For positively-worded items (odd numbers), the score contribution is the scale position minus 1. For negatively-worded items (even numbers), the score contribution is 5 minus the scale position. To get the overall SUS score and obtain a range from 0 (very poor perceived usability) to 100 (excellent perceived usability) in 2.5-point increments as explained in [20], the sum of the item score contributions is multiplied by 2.5:
(1)$$\begin{array}{c} X=\mathrm{Sum}\;\mathrm{of}\;\mathrm{the}\;\mathrm{points}\;\mathrm{for}\;\mathrm{all}\;\mathrm{odd-numbered}\;\mathrm{questions}-5\\ Y=25-\mathrm{Sum}\;\mathrm{of}\;\mathrm{the}\;\mathrm{points}\;\mathrm{for}\;\mathrm{all}\;\mathrm{even-numbered}\;\mathrm{questions}\\ \mathrm{SUS}\;\mathrm{Score}=(X+Y)\cdot 2.5 \end{array}$$

In our study, Equation (Equation 1) is used to analyse the SUS for each of the involved participants, while the obtained score is analysed as previously described and also illustrated in Figure 4 according to Usability.gov (https://measuringu.com/interpret-sus-score/, accessed on 1 January 2021) which a SUS score above 68 is considered average, and anything lower than 68 is below average.

## 3. Results

### 3.1. Participants

The participants (n = 11) involved in this study were called novices as they had never used a wheelchair before. Pseudonyms were used to protect participants’ anonymity. The participants’ age ranged from 22 to 38 (only one participant was female). All participants realised the three experimental steps, e.g., familiarisation with the IPW, running the predefined path and filling out the questionnaire, in less than 60 min (range: 45–60 min).

### 3.2. Findings

Some overarching themes resulted from our qualitative and quantitative data analysis, represented by (i) the participant observation method, (ii) the SUS questionnaire and the four open-ended questions related to the IPW functionality.

#### 3.2.1. Qualitative Data Analyses: Participant Observation

During the participant observation stage, two categories of participants were observed during the familiarisation step: (a) 9 out of 11 participants tested each mode one by one in order to understand how to use them, (b) while the others 2 participants tested each of the control modes in a random way. Generally, all the participants were very autonomous in performing the indicated steps, with a desire of understanding the system by themselves without asking too many questions. Still, 8 out of 11 participants asked many pertinent questions related to the modules attached to the IPW, the limitations of the system or why the systems reacted in a certain way. For example, 2 out of 11 participants had problems during the trajectory, as they tried to push the system’s limit, such as going too close to the obstacle or increasing the IPW’s speed too much. Even so, they did not abandon the experiment and performed all the steps. They rather tried performing the scenarios involved in the experiment in order to understand if the problem came from the IPW or from their way of performing the trajectory. Still, one participant decided to not use the IPW-shape-joystick, as from his perspective it was not so easy to use. While ten out of eleven participants realised only three times the predefined trajectory, either using each control mode one after the other or mixing the control modes, only one participant performed the predefined trajectory more than three times, trying to use each mode as well as mixing the modes to follow the path.

Regarding the software used for the proposed graphical interface, ROS (https://www.ros.org/about-ros/, accessed on 1 January 2021 ), 2 out of 11 novice participants already made use of it previously in their career. For this reason, they really tried to put themselves in an IPW user’s place. Moreover, at the same time they raised good questions and offered critical feedback on the proposed interface.

It was also interesting to see that even if 2 out of 11 participants were very tall and the touch screen could not be adjusted to a good position for them, they still went through the experiment steps and provided exploitable and good feedback.

#### 3.2.2. Quantitative Data Analyses I: SUS Questionnaire

Table 1 shows the score contributions reported by each participant to the SUS questionnaire. The odds-numbered items, e.g., frequent usability (**SUS1**), perceived ease of use (**SUS3**), perceived system integration (**SUS5**), difficulty to learn (**SUS7**), confidence in use (**SUS9**) should have a positive tone, while the even-numbered ones, e.g., perceived complexity (**SUS2**), technical need (**SUS4**), perceived consistency (**SUS6**), perceived awkwardness/cumbersomeness/heaviness (**SUS8**), perceived learnability (**SUS10**) usually have a more negative tone.

**SUS1** (frequent usability): 1/11 participants responded in a neutral way, 3/11 participants responded in a positive way thereby following the instruction provided by the observer, while the other 8 participants responded in a negative tone, thereby going the other way around.**SUS2** (perceived complexity): 3/11 responded with a neutral tone while the rest of the participants had a negative one.**SUS3** (perceived ease of use): 2/11 participants responded in a neutral way, while the rest of 9 participants responded in a positive tone.**SUS4** (technical need): only 1/11 participant responded in a positive tone, while the others provided a negative response.**SUS5** (perceived system integration): only 1/11 participant responded in a negative tone, 2/11 participants used a neutral tone, while the other ones provided a positive one.**SUS6** (perceived consistency): 3/11 used a neutral tone, 1/11 had a positive tone, while the rest indicated negative ones.**SUS7** (difficult to learn): 2/11 remained neutral 1/11 participant responded in a negative tone, and the other ones provided a positive one.**SUS8** (perceived awkwardness/cumbersomeness/heaviness): 1/11 participant responded in a neutral way, while the other ones provided a negative one.**SUS9** (confidence in use): 2/11 had negative tones, 3/11 remained neutral, while the others indicated positive tones.**SUS10** (perceived learnability): only 1/11 participant responded in a positive tone, while the other ones provided a negative one.

Each participant’s choices in the SUS questionnaire are as follows:**P1** remains neutral to using frequently the system (**SUS1**), to the perceived ease of use (**SUS3**) and the need for technical support (**SUS5**), while answering in the expected way to other questions.**P2**: indicates neutral response to perceiving the system as being complex (**SUS2**), to the perceived ease of use (**SUS3**) and to the perceived consistency (**SUS7**); while not feeling too confident in using it (**SUS9**), would not frequently use the system (**SUS1**) and needs to learn a lot of things before using it (**SUS10**);**P3**: would not use frequently the system (**SUS1**) and remains neutral to perceiving the system as being complex (**SUS2**) and there being too much inconsistency with the system (**SUS6**);**P5**: indicated that they would not use frequently the system (**SUS1**), found that the system would not be learned too quickly by others (**SUS7**), have not found confidence in the system (**SUS9**), perceived system integration (**SUS4**), though there was too much inconsistency with the system (**SUS6**), while being neutral to founding the system very cumbersome to use (**SUS8**) and the need for technical support (**SUS5**);**P7**: would not use frequently the system (**SUS1**) and remained neutral regarding the confidence in the system (**SUS9**).**P10**: indicated that the system does not have the features well integrated (**SUS5**) and does not feel too confident in using it (**SUS9**); while remaining neutral to perceiving the system as being complex (**SUS2**), to having too much inconsistency (**SUS6**) and found that the system would not be learned too quickly by others (**SUS7**). This participant is the only one who indicated that as a wheelchair user he would frequently use the SmartWheeler.**P11**: indicated that he would not use frequently the system (**SUS1**), thought there was too much inconsistency with the system (**SUS6**) and does not feel to confident in using it (**SUS9**)**P4**, **P6**, **P8**, **P9**: in all but one aspect, the frequency usability of the system (**SUS1**) to which **P4** remained neutral, the others indicated a negative tone, and answered as expected.

The SUS scores and grades reported in Table 2 and graphically represented on Figure 5, found that the participants can be classified in two (2) groups:Group 1: 4 out of 11 participants scored the SUS below the average of 68, meaning they consider the IPW as a poor system or at the limit of being an ok one;Group 2: 7 out of 11 participants scored the SUS above or equal to the average of 68, which indicates that they consider the IPW system as an *ok*, *good* and *excellent* system.

**Figure 5 sensors-22-05627-f005:**
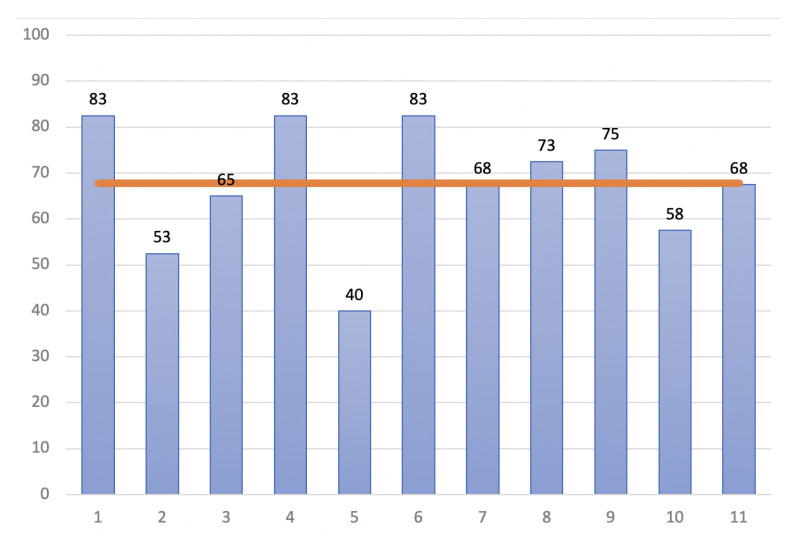
Graphical representation of the SUS scores, for the odds items, even items and grades obtained for each participant along the average one (represented by the orange line).

The reported results exhibited from our SUS questionnaire while analysing the data are as expected, meaning that the proposed IPW has an *ok* usability, a *marginal* acceptability and a *passive* net promotor score (NPS) and a 50% percentile rank (as Figure 4 suggests).

#### 3.2.3. Quantitative Data Analyses II: The Four Open-Ended Questions, Related to IPW Functionality

The collected data of the four open-ended questions (reported in Section A.2) related to IPW functionality are analysed in this section. The results of the first three questions are grouped as *Choice, advantages and disadvantages* and question the choice of using a specific operation mode, the advantages and the disadvantages of such operation modes.

**Q**${}_{1}$.For what reason(s) did you choose to activate/use one mode of operation rather than another at a specific point in the journey?**Q**${}_{2}$.What advantage(s) do you find in each of the three modes?**Q**${}_{3}$.What disadvantage(s) do you find in each of the modes?

These three questions are reported in the first part of this subsection. The following reports the feedback provided by the participants in terms of *proposed improvements* for the IPW:**Q**${}_{4}$.What improvements to the IPW would you suggest?

The proposed questions were identified by the authors as an extension to the SUS questionnaire. As the SUS questionnaire gives answers to the overall "system" such as its performances and all its functionalities, these four questions allowed the participants to provide feedback regarding the interface in use and to propose improvements.

##### Choice, Advantages and Disadvantages

The data collected for this group were analysed using constructors presented within The Consolidated Framework for Implementation Research (CFIR) [21]. Each identified constructor was rated using the **valence** rating, which consists in identifying two factors: *positive and negative influence on the system and interface*. First the provided comments to each question were classified using the valence rating and reported in tables. Then, common comments are reported.

### 3.3. Manual, Semi-Autonomous and Autonomous Mode

The feedback represented by participants’ comments is reported in the following. The collected comments are grouped in function of the variance rating *positive*, *negative* and *neutral* or *no answer* when there is no comment. The comments provided by the novice participants to the first group of questions regarding the choice, advantages and disadvantages of each control mode are reported in Table 3, Table 4 and Table 5 as follows:Table 3 gathers comment for manual mode.-(Choice of the mode) Two out of eleven participants reports negative feedback: *Gives a false confidence linked to a habit* and *Acceleration was fast*. These two comments are linked as the first one suggests that the use of a joystick can be intuitive but while activated the IPW can move very fast.-(Advantages) Only one participant out of eleven does not see an advantage in using the manual mode.-(Disadvantages) Two out of eleven participants do not see a disadvantage in using the manual mode.Table 4 gathers comment for the semi-autonomous mode.-(Choice of the mode) Five out of eleven participants gave negative comments regarding the choice to use the IPW-shape-joystick, while nine out of eleven found the semi-autonomous mode *intuitive*, *easy to use*.-(Advantages) One participant indicates negative feedback, but the same participant also provides positive feedback as well. Moreover, P${}_{3}$ was neutral by indicating *I do not know*, while P${}_{4}$ does not provide an answer to this question.-(Disadvantages) Nine out of eleven indicate some disadvantages in using this mode but more precisely the IPW-shape-joystick one. P${}_{9}$ prefers to not answer the question, while P${}_{3}$ provides an advantage rather than a disadvantage *It is advisable to use the virtual joystick*.Table 5 gathers comment for the autonomous mode.-(Choice of the mode) Five out of eleven participants report negative comments regarding their choice of using such mode. Still, all the comments are related to the choice our team made in leaving the IPW to choose the path to follow and not giving the participant the choice to select one. P${}_{3}$ remains neutral, while P${}_{6}$ preferred not to answer.-(Advantages) Only one participant, P${}_{1}$, did not report any advantage in using the autonomous mode.-(Disadvantages) P${}_{9}$ preferred to not answer, while P${}_{1}$ already reported its disadvantages in the previous questions. The rest of the participants found disadvantages related once again to the choice of our team to leave the IPW to choose the path to follow and the final orientation without giving the participant control over it.

### 3.4. Proposed Improvements

The improvements provided by the participants are divided into two areas: (a) *IPW software* which will cover all feedback regarding the possible improvements to the interfaces/software available on the IPW and (b) *IPW hardware* where we considered all feedback regarding the improvements related to the system control and behaviour. The used rating is based on the **strength** (e.g., weak or/and strong) influence on implementation [21]. First, the improvements regarding the IPW software and IPW hardware are reported in the order collected from $P_{1}$ to $P_{11}$, as follows:

#### 3.4.1. IPW Software

More ergonomic interface (especially for the IPW-shape-joystick);To have the possibility to better define the final positioning (autonomous mode);Having the ability to bypass obstacle detection;Have the possibility to increase the communication between the IPW and the user if needed (the IPW can use the human to confirm some data or hypothesis);Improve the final angular position (make something more intuitive IPW-shape-joystick);Continue to develop the performance of the autonomous mode (Pay attention to orientation);The map with the vector cloud becomes confusing and does not help navigation when there are too many points;Possibility to position the IPW in a certain direction at the end of a maneuver in autonomous mode;Be able to set a desired orientation in the standalone interface;Use of the manual joystick for the semi-autonomous mode;Prioritization of the manual over the autonomous;Reduction of the radius of obstacles;Display the trajectory of the IPW in autonomous mode;Decide the arrival point of the IPW in autonomous mode;Display of the object that blocks the semi-automatic and automatic modes;Clearer selection of which mode I am currently using.

#### 3.4.2. IPW Hardware

Improve control over rotation (faster rotation);Better motor response (progressive acceleration curve);Revising the controls backwards (Reverse according to my impression, but I am not a IPW user) (Semi autonomous and Manual);Add tactile feedback on the controls (Semi autonomous and autonomous): vibration maybe?;Make the controls more fluid;Find ways that the user doesn’t have to look at the tablet during development;I propose to add a control of the system on the smart phone (it’s more pleasant) otherwise it’s good that the position of the tablet is adjustable (to be compatible with different sizes);I also propose to add rear cameras (to have a 360 vision, and know what happens around us);Better control of the speed of the IPW in semi-autonomous and autonomous mode;Maybe add a rear camera (easier than turning around);Height adjustment on the touch screen, I would find it relevant to have a physical emergency stop, the one on the touch screen gives me less confidence;Fluidity in the control would be appreciated in my opinion;Replacement of the manual joystick for a larger less sensitive one;More intuitive reversing;Smoother control of the motors during turns;Use of physical buttons.

Table 6 groups the common comments for both IPW software and hardware. Moreover, a strong or/and weak ranking is associated with each common comment. From the common comments regarding the *IPW software* it can be seen that (i) the autonomous mode was really appreciated, however, there is a requirement to involve the users more in the decision making and (ii) the graphical interface should be more user-friendly, which we already have. Both comments were expected to emerge due to the choices made by our team to use an open-source graphical interface library usually used by the robotics community and by not giving the choice of path to follow and orientation for the autonomous mode. The suggested improvements for the *IPW hardware* are appealing as the participants require improvements in order to increase the confidence they put in the wheelchair, such as *Add vibrations feedback on the controls*, *Use of physical buttons for emergencies* or *Find ways to not look at the tactile screen*.

Overall, all the suggested improvements are consistent and are taken into account by our team.

## 4. Discussion

This qualitative and quantitative study is the first of its kind to explore the usability and functionality of an IPW, developed by our team, by novice participants without any experience in driving one. Previous studies done with the same IPW to explore the perceptions of IPW with end-users, via interviews before and after testing the IPW [6] or via interviews after illustrating video [5] with the IPW exhibited positive results and suggests that IPW would enhance social participation in a variety of important ways. This motivated us to continue developing the IPW until it can be capable of being fully usable by the intended users. Thereby, the feedback provided by novice participants who had a critical eye on the used technology helps us (a) adjust the physical IPW and the proposed graphical interface and (b) go further and perform tests in real environments with IPW users. The fact that novice users found the IPW acceptably usable with good functionalities suggests that the IPW can be adjusted by taking into account the feedback provided in this study, and could be used to improve user’s mobility and lifestyle.

Still, along with positive feedback, the participants also raised technological and personal concerns, such as

the IPW’s touch screen not being adaptable for tall participants;the proposed graphical interface which proposes different control modes should be more user-friendly;the autonomous navigation, even if it was very appreciated, should be improved;the map used on the graphical interface should be more realistic;the manual joystick, as well as other control modes of the IPW, should be thought for left and right-handed users.

Going further, the feedback and results collected through this study will allow us to improve the IPW before testing it with the users in real environments. It is also worth mentioning that the current word pandemic situation requires additional features to the IPW, such as (i) human-aware navigation, which makes the IPW capable of interacting with its environment (with the already existing autonomous mode) and taking decisions based on the human social interaction existing in it and can ensure a safe distance between the user and other humans; (ii) modules allowing the IPW to follow a person thus give the users the ability to “walk” alongside their caregivers, family or someone from the community while keeping a safe distance between them. These ideas already have received attention in the literature. For example, the socially acceptable navigation or human-aware navigation to socially interact with people [22] or to learned cost functions to be used for social navigation [23] are already studies which report positive outcomes prior to the current word pandemic. A review of sociological concepts of social robotics and human-aware navigation is proposed in [24]. We believe we can go further and use socially acceptable collision avoidance in an outdoor environment as the one proposed in [25] for a mobile robot, as the IPW can be considered a mobile robot and thus easily adopt such approaches. Additionally, ramp detection models can be implemented using deep learning [26], which could improve the mobility of the user, as well as modules which enable side-by-side following [27], which could make IPW users interact more with their caregivers instead of focusing on navigating the IPW. One can also find studies which propose low-cost and robust real-time eye-controlled wheelchair [28], user’s voice [29], facial expression [30] and brain-controlled wheelchair to navigate in familiar environments with healthy [31] or patients [32] are proposed as solutions for people losing their sensing ability. Currently, our team is working on developing a low-cost BCI-based graphical interface in order to assist the control of the IPW, add human-aware navigation in the already existing navigation modes and use new technologies to allow the user to take a walk with a specific person indoors and outdoors.

## 5. Limitations

The reported study has some limitations. First of all, the use of novice participants can be considered a limitation of this study, as it is difficult to project in the daily needs required by an IPW end-user. Furthermore, even if the data analysis for the SUS questionnaire resulted in a positive outcome, it is worth underling the fact that the SUS questionnaire is intended for IPW users. Therefore, a bias can be exhibited in the first question, e.g., *I think that I would like to use this feature frequently*. However, each participant was instructed verbally by the observer and also indicated in writing, e.g., “*To answer the questions, imagine yourself as a motorized wheelchair user*”, at the beginning of the questionnaire”.

Secondly, even if done purposefully and knowingly, the use of a virtual map, generated by the Rviz graphical interface directly on the touchscreen did not give a good sense of spatial orientation to the users. For this reason, such input from the users was not a surprise. Moreover, future work will make use of a map which will better represent the environment in which the IPW navigates.

Moreover, the participants found that the autonomous mode, which gives the IPW the ability to navigate by itself from one destination to another, takes time to perform the trajectory and to stabilise. This is due to the fact that our team decided to use methods which allow the IPW to decide on the chosen path to the destination at a lower speed in order to avoid fearful reactions from the participant’s side.

Finally, even if the novice users did not have experience driving an IPW prior to this experiment, the collected data exhibited that there were two participants, P${}_{2}$ and P${}_{5}$, which associated the manoeuvre of the manual joystick with similar systems such as *video games*. Therefore, there may be a bias for these two participants, as they will prefer something which they already used. However, as regarding their feedback for the other modes, the authors did not find such a bias as their comments were coherent as reported on Table 4 for the semi-autonomous mode and Table 5 for the autonomous one.

## 6. Conclusions

In this study, eleven novice participants, with and without knowledge of the technology used to improve the IPW, agreed to participate in our qualitative and quantitative study. The improved IPW was first introduced within the SmartWheeler project.

The quantitative analyses suggest that the IPW is sufficiently usable and robust. Moreover, the qualitative analyses via the participant observation method allowed us to learn how to further form end-users at using the proposed control modes. Finally, the second quantitative analyses method helped us (i) uncover issues which we need to address before using it with end-users, for example distinguishing between left- and right-handed persons and (ii) understanding what are the first impressions on the proposed control and navigation modes.

Future work will involve testing improvements to our IPW prototype based on the feedback and insights from this study with the aim of enhancing mobility and quality of life of IPW end-users with and without loss of the sense of touch.

## Figures and Tables

**Figure 3 sensors-22-05627-f003:**
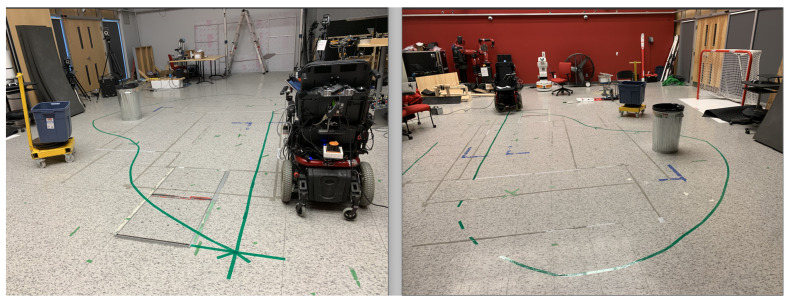
The predefined path marked by green tape on the floor of an experimental room of the 3IT at Université de Sherbrooke.

**Figure 4 sensors-22-05627-f004:**
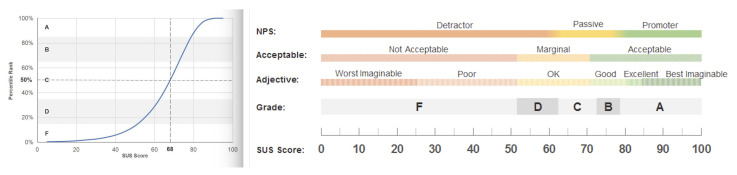
SUS on a curve with percentile ranks and grades (Usability.gov, accessed on 1 January 2021).

**Table 1 sensors-22-05627-t001:** Answers provided by each participant to the SUS questions.

	P1	P2	P3	P4	P5	P6	P7	P8	P9	P10	P11
SUS1	3	1	1	3	1	1	1	1	1	4	1
SUS2	1	3	3	2	2	1	1	2	2	3	2
SUS3	3	3	4	4	4	5	4	4	4	4	4
SUS4	1	2	1	1	5	2	2	2	2	2	1
SUS5	3	4	4	4	3	4	4	4	4	2	4
SUS6	1	2	3	1	4	2	2	1	2	3	3
SUS7	4	3	4	4	2	5	4	4	5	3	4
SUS8	1	2	2	1	3	1	2	2	1	2	2
SUS9	5	3	4	4	2	5	3	4	4	2	3
SUS10	1	4	2	1	2	1	2	1	1	2	1

**Table 2 sensors-22-05627-t002:** Tabular representation of the SUS scores, for the odds items, even items and grades (as explained in Figure 4) obtained for each participant along the average one.

System Usability Scale (SUS) Score			
**Odd Items**	**Even Items**	**SUS Score (/100)**	**Grades**
13	20	83	A
9	12	53	D
12	14	65	D
14	19	83	A
7	9	40	F
15	18	83	A
11	16	68	D
12	17	73	B
13	17	75	B
10	13	58	D
11	16	68	D
Average SUS Score		68	D

**Table 3 sensors-22-05627-t003:** Manual mode.

	Feedback	Participants	Comments
Q${}_{1}$. Choice	Positive	P${}_{1}$, P${}_{2}$, P${}_{3}$, P${}_{4}$, P${}_{5}$, P${}_{6}$, P${}_{7}$, P${}_{8}$, P${}_{9}$, P${}_{10}$, P${}_{11}$	*Not affected by the obstacle’s presence*; *A more natural method due to video games* *More control in the rotation motions* *More instinctive...* *...when you feel the need to look around* *Quick reaction and easy to control, very intuitive* *I feel I have more control over the system* *To have more precision and to pass closer to the...* *...obstacles or when the way shrinks* *The most intuitive and easy to control mode*; *Allows you to react as quickly...* *...and accurately as possible*; *Used for making small adjustments* *Test and gain confidence in the IPW* *To pass in the straight places...* *...and to direct freely without constraint* *Easy to control* *To test the sensibility of the IPW*
	Negative	P${}_{4}$, P${}_{11}$	*Gives a false confidence linked to a habit* *Acceleration was fast*
Q${}_{2}$. Advantages	Positive	P${}_{1}$, P${}_{2}$, P${}_{3}$, P${}_{4}$, P${}_{5}$, P${}_{6}$, P${}_{7}$, P${}_{8}$, P${}_{9}$, P${}_{10}$, P${}_{11}$	*easy to use* *more common, natural,..* *...more used to use a similar system (video games),...* *...easier to understand* *better control and greater speed* *More instinctive...* *...when you feel the need to look around* *simple and efficient* *very easy to use...* *...and understand (like a game joystick)* *move faster and avoid obstacles* *easy, intuitive, precise* *makes you feel more in control* *no constraints, I have my autonomy* *more reassuring*; *allows for more complex maneuvers*
	Negative	P${}_{4}$	*Gives a false confidence linked to a habit*
Q${}_{3}$. Disadvantages	Positive	P${}_{1}$, P${}_{8}$	*none* *I don’t really see any*
	Negative	P${}_{2}$, P${}_{3}$, P${}_{4}$, P${}_{5}$, P${}_{7}$, P${}_{9}$, P${}_{10}$, P${}_{11}$	*you have to learn how to drive* *difficult to use with the left hand* *requires constant concentration* *request more dexterity* *need for more fluidity and reactivity* *I can hit an obstacle at full speed and cause a fall* *I always have to look around*
	No answer	P${}_{6}$	

**Table 4 sensors-22-05627-t004:** Semi-autonomous mode.

	Feedback	Participants	Comments
Q${}_{1}$. Choice	Positive	P${}_{1}$, P${}_{3}$, P${}_{4}$, P${}_{6}$, P${}_{7}$, P${}_{8}$, P${}_{9}$, P${}_{10}$, P${}_{11}$	*The virtual joystick is easy to use* *When moving straight ahead* *Gives excellent visual feedback* *Even if the joystick was on the left,...* *...I’m right-handed,...* *...the joystick was nice to use* *The easiest and most intuitive mode for me...* *...because you can easily readjust the system...* *...if it does something unexpected* *This mode is also quite intuitive* *I guess I would use this mode...* *...if I don’t trust my movements* *Allows to see the obstacles*
	Negative	P${}_{1}$, P${}_{2}$, P${}_{4}$, P${}_{5}$, P${}_{9}$	*The IPW-shape-joystick is very hard to use* *Using the tablet is more complicated...* *...than the manual joystick* *I might prefer if I don’t try to take...* *...my eyes off my screen* *I would see no reason to use...* *...it if I could use the manual joystick* *The IPW-shape-joystick was the most difficult to...* *...control the speed of movement...* *...as well as the turn*
Q${}_{2}$. Advantages	Positive	P${}_{1}$, P${}_{2}$, P${}_{5}$, P${}_{6}$, P${}_{7}$, P${}_{8}$, P${}_{9}$, P${}_{10}$, P${}_{11}$	*Easy to use the virtual joystick* *Seems to have a greater sensitivity and...* *...precision on the position of the joystick* *Possible to use it without dexterity...* *...care on the hands* *It is very easy to use and understand* *More precise than the manual joystick...* *...and slower speed so you can readjust...* *... more easily during the course...* *...and derive from where you want to go* *Intuitive* *I protect the environment...* *...and the people around me* *Avoid the obstacles*
	Negative	P${}_{1}$	*Difficult to use the IPW-shape-joystick*
	Neutral	P${}_{3}$	*I do not know*
	No answer	P${}_{4}$	
Q${}_{3}$. Disadvantages	Positive	P${}_{8}$, P${}_{3}$	*Desirable for visual representation* *Rarely blocks* *The IPW moves a little faster* *It is advisable to use the virtual joystick*
	Negative	P${}_{1}$, P${}_{2}$, P${}_{4}$, P${}_{5}$, P${}_{6}$, P${}_{7}$, P${}_{8}$, P${}_{10}$, P${}_{11}$	*The IPW-shape-joystick is very difficult to understand* *Less pleasant to use than a manual joystick* *We must always be aware of where we want to go* *Requires increased concentration on the screen* *It is difficult to predict the movement (IPW-shape-joystick)* *Speed control is more difficult* *The IPW-shape-joystick is more difficult to use*
	No answer	P${}_{9}$	

**Table 5 sensors-22-05627-t005:** Autonomous mode.

Mode 3	Feedback	Participants	Comments
Q${}_{1}$. Choice	Positive	P${}_{1}$, P${}_{2}$, P${}_{4}$, P${}_{7}$, P${}_{8}$, P${}_{9}$, P${}_{10}$, P${}_{11}$,	*Easy to use* *Interesting to not have to think about moving...* *...and just ask to move to a place* *To travel long distances without concentrating*; *It is extremely pleasant to move in an autonomous way* *Is the most fun fashion of all*; *This is the mode I preferred* *It was interesting to be driven to a particular point,...* *...I like the autonomous route...* *...I would use it for long distances* *If I had a big handicap I could get around without help* *Easy to use*; *Pleasant*
	Negative	P${}_{1}$, P${}_{2}$, P${}_{4}$, P${}_{5}$, P${}_{7}$	*The selection mode of the arrival point is not intuitive* *Impossible to define the position in rotation* *The random movement and did not give confidence* *In a real situation I would avoid using it* *A little more difficult to use because of...* *...the presence of obstacles in the form of clouds* *The systems takes a lot of time, however*
	Neutral	P${}_{3}$	*I do not know*
	No answer	P${}_{6}$	
Q${}_{2}$. Advantages	Positive	P${}_{2}$, P${}_{3}$, P${}_{2}$, P${}_{5}$, P${}_{6}$, P${}_{7}$, P${}_{8}$, P${}_{9}$, P${}_{10}$, P${}_{11}$	*Very simple to make the IPW understand...* *...where I want to go* *To move to a known house* *No action to do on the tablet to give a place to move* *It’s easy to use and understand* *The easiest way to get to a specific point* *Less action on my part*; *More fun* *Good for getting to a destination further away,...* *...a map with multiple floors would be nice with this* *I don’t have to control anything,...* *...I am guided by the IPW* *No need to think, nice to be driven around*
	Negative	P${}_{1}$	*For the moment none* *Less simple to tell the IPW which path is better* *The tolerance on the stop point could be bigger*
	No answer	P${}_{4}$	
Q${}_{3}$. Disadvantages	Negative	P${}_{2}$, P${}_{3}$, P${}_{4}$, P${}_{5}$, P${}_{6}$, P${}_{7}$, P${}_{8}$, P${}_{10}$, P${}_{11}$	*Difficult to define the destination point; Not intuitive* *To make small trips, it quickly becomes more complex* *Angular position difficult to control* *No sense of control*; *Very random movement* *Freezes; Lacks steering control* *Difficult to pass if there are several obstacles...* *...close to each other, I need to use another mode* *Too bad we can’t specify the final orientation* *Takes a long time to stabilize* *I have no control over my movements*; *I can’t adjust my angle of arrival* *The use of a touch screen*; *Unnatural acceleration*; *The IPW took time to stabilize at the destination*
	No answer	P${}_{9}$	

**Table 6 sensors-22-05627-t006:** Common areas of improvements along with the common improvement and the allocated ranking score.

Area of Improvement	Common Comment	Ranking
IPW software	More user-friendly interface	strong
	Improve destination selection	strong
	Better obstacle display	weak
	More human involvement with the IPW	weak
	Continue autonomous mode development	strong
	Prioritize the manual joystick over other modes	weak
IPW hardware	Fluidity in the motor control	strong
	Rethink the reverse control	weak
	Add vibrations feedback on the controls	strong
	Find ways to not look at the tablet	strong/weak
	Use a smartphone	weak
	Height adjustment on the touch screen	strong
	Larger less sensitive joystick	weak
	Use of physical buttons for emergencies	strong

## Data Availability

Not applicable.

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
