# Peer review of "Usability Evaluation of the SmartWheeler through Qualitative and Quantitative Studies"

_sensors, 2022, doi:10.3390/s22155627_

Round 1

Reviewer 1 Report

This paper uses qualitative and quantitative methods to test the usability of IPW with more details that are commendable.

However, the technical contribution is few. Besides, there are numerous concerns to be addressed in the manuscript.

In the abstract, you claimed, "there are still open issues regarding the usability and functionalities of an Intelligent Powered Wheelchair (IPW)," so your aim in this thesis is not to improve these usability issues but to re-run usability tests?

Are the qualitative and quantitative studies written backward in the summary results?

The qualitative analysis done with novice users reports an "okay" rating (equivalent to a C grade or 68 SUS Score) for our IPW usability. Moreover, the quantitative analysis helped gathered advantages and disadvantages opinions on the IPW but also comments which can be used to improve the system.

Why choose the wheelchair with a display? There are many smarter wheelchairs available, such as those that use EEG/EOG? What were the reasons for your choice? What are its advantages over other wheelchairs? Take a look at reference [13]; why choose it? Isn't it just too long ago?

I can't quite see the contribution of this article; as you say in line 63, testing IPW usability is not new, so where is the innovation in your test or paper?

The format of references 13 and 18 appears to be incorrect.

Most of the references are outdated.

To conclude, I am afraid I cannot vote positively for this work's publication.

Reviewer 2 Report

Thanks for recommending me as a reviewer. In this paper, authors presents an experience with our IPW followed by a study in two parts: a quantitative one based on the System Usability Scale  questionnaire and a qualitative one through open questions regarding IPW functionalities with novice users. These users never used an IPW before, but are users and aware of the impacts of the technology used in our IPW, being undergraduate to postdoctoral students and staff at the Faculty of Engineering of Université de Sherbrooke. The qualitative analysis done with novice users reports an "okay" rating for our IPW usability. If authors complete minor revisions, the quality of the study will be further improved.

1. The introduction section is well written. But it's too verbose. If the authors describe the introduction section more clearly, it can help readers understand it.

2. Is Figure 1 necessary for this study? If the authors describe the characteristics of Figure 1 more specifically, it may help the reader's understanding.

3. If the authors describe the conclusion section more specifically, it may help readers understand.

Reviewer 3 Report

Dear authors,

Your article addresses a relevant and interesting topic. Thank you for the opportunity to read your manuscript.

The following are some suggestions that I hope will help improve the text. I believe the text needs to be reorganized to make it more attractive.

The manuscript is too long for today's readership. Some information is repeated unnecessarily. Other important ones are unclear. The objectives, for example, are subdivided into four and it is not possible to analyze whether the conclusion answers all.

The presentation of results could be rethought. You have many results and presenting all of them makes it difficult for the reader to synthesize them.

I believe the limitations could be presented at the end of the discussion.

The fact of using people without disabilities to evaluate the system needs to be reviewed. I believe that people with experience in using a wheelchair could provide more important answers for the study.

Reviewer 4 Report

Authors describe results of their research on innovative solutions for existing   Intelligent powered wheelchair utilised already in an another project - the SmartWheeler project. The proposed new functions were experimentally tested with 11 "novice" users (non users of IPW before), and results evaluated using qualitative and quantitative methods, Methodically, the research was well planned and done.  However, authors know that their research with promising good results needs improvements in several issues like to distinguish set-up of navigation for left- and right handed persons, improve graphical interface, etc. Currently, they are working on developing a low cost BCI-based graphical interface in order to assist the control of the IPW, add human-aware navigation in the already existing navigation modes and use new technologies to allow the user to take a walk with a specific person indoor and outdoor. They want to continue in the research with the aim of enhancing mobility and quality of life of IPW end-users with and without loss of touch sense.  In my opinion such goals might significantly contribute to improvements of  IPW functionalities.

Authors used 34 references. There is incorrect description of the following references: 10, 13, 18, 34. It is necessary to complete identification data for them. 

Round 2

Reviewer 1 Report

In this paper, the usability of a specific mode of operation of a powered wheelchair, i.e. the virtual joystick coupled with an interactive display, is tested by quantitative and qualitative methods. The experimental method, the experimental procedure, and the results are clearly described. It is a useful reference for similar tests on IPWs.